

# Teleost and elasmobranch eye lenses as a target for life-history stable isotope analyses

Katie Quaeck-Davies[1,*], Victoria A. Bendall[2], Kirsteen M. MacKenzie[3], Stuart Hetherington[2], Jason Newton[4] and Clive N. Trueman[1,*]

[1] Ocean and Earth Science, University of Southampton, Southampton, United Kingdom
[2] Centre for Environment, Fisheries and Aquaculture Science, Lowestoft, United Kingdom
[3] Institute of Marine Research/Havforskningsinstituttet, Bergen, Norway
[4] Scottish Universities Environmental Research Centre, University of Glasgow, Glasgow, United Kingdom
[*] These authors contributed equally to this work.

Corresponding author
Clive N. Trueman,
trueman@noc.soton.ac.uk

## ABSTRACT

Incrementally grown, metabolically inert tissues such as fish otoliths provide biochemical records that can used to infer behavior and physiology throughout the lifetime of the individual. Organic tissues are particularly useful as the stable isotope composition of the organic component can provide information about diet, trophic level and location. Unfortunately, inert, incrementally grown organic tissues are relatively uncommon. The vertebrate eye lens, however, is formed via sequential deposition of protein-filled fiber cells, which are subsequently metabolically inert. Lenses therefore have the potential to serve as biochemical data recorders capturing life-long variations in dietary and spatial ecology. Here we review the state of knowledge regarding the structure and formation of fish eye lenses in the context of using lens tissue for retrospective isotopic analysis. We discuss the relationship between eye lens diameter and body size, describe the successful recovery of expected isotopic gradients throughout ontogeny and between species, and quantify the isotopic offset between lens protein and white muscle tissue. We show that fish eye lens protein is an attractive host for recovery of stable isotope life histories, particularly for juvenile life stages, and especially in elasmobranchs lacking otoliths, but interpretation of lens-based records is complicated by species-specific uncertainties associated with lens growth rates.

## INTRODUCTION

Retrospective chemical analyses of inert, incrementally formed tissues provide valuable insights into (for instance) the trophic and spatial ecology of animals (i.e., their trophic geography) (*Best & Schell, 1996*; *Bird et al., 2018*), population connectivity and stock structure (*Campana, 1999*) and interactions between animals and climate (*Trueman, MacKenzie & Palmer, 2012*). This source of information can be particularly useful for elusive marine animals where direct observation and physical tagging is complicated by large spatial distances and inaccessible habitats.

Incrementally grown tissues are commonly used for retrospective biochemical assessment (*Tzadik et al., 2017*). The trace element and stable isotope chemistry of otolith aragonite has been used extensively to provide a chronological chemical record of the water inhabited by the fish (e.g., *Kalish, 1991*; *Thorrold et al., 1997*; *Campana, 1999*; *Høie, Otterlei & Folkvord, 2004*; *Tzadik et al., 2017*), addressing questions such as stock mixing and nursery origin of Atlantic bluefin tuna (*Thunnus thynnus*) (*Schloesser et al., 2010*), ontogenetic depth migration in the roundnose grenadier (*Coryphaenoides rupestris*) (*Longmore et al., 2011*) and migration in Atlantic salmon (*Salmo salar*) (*Hanson et al., 2010*). However such inorganic records do not provide information on trophic ecology. Recently, isotopic analyses of the organic component of otoliths (*Grønkjær et al., 2013*) open the potential for retrospective reconstruction of ontogenetic variation in feeding and spatial ecology of individual fishes, but the sample sizes required currently limit the application of the method. The isotopic composition of carbon and nitrogen in incrementally grown organic tissues such as baleen (*Best & Schell, 1996*; *Hobson & Schell, 1998*; *Lee et al., 2005*), vertebral collagen (*Estrada et al., 2006*; *Kim et al., 2012*) and vibrissae (*Cherel et al., 2009*) is particularly attractive as an ecological tracer (*Tzadik et al., 2017*). Variations in stable isotope compositions of carbon, nitrogen and sulfur within incrementally-grown tissues can be interpreted with respect to changes in diet or trophic level and/or variation in the isotopic composition of primary production across areas or habitats (the isotopic baseline).

The vertebrate eye lens is another incrementally grown, metabolically inert, proteinaceous tissue and as such is potentially suitable for retrospective recovery of life history biochemical data (*Dove & Kingsford, 1998*; *Kingsford & Gillanders, 2000*; *Gillanders, 2001*; *Parry, 2003*; *Wallace, Hollander & Peebles, 2014*; *Tzadik et al., 2017*). The vertebrate eye lens has received comparatively little attention as a host for life history biochemical data (*Parry, 2003*; *Wallace, Hollander & Peebles, 2014*). Effective use of the eye lens as a tissue for hosting isotope data depends on knowledge of the structure, growth and isotopic systematics of lens proteins. In this study we briefly review the structure and growth of fish eye lenses, test the performance of lens-hosted stable isotope compositions to recover known or expected isotopic contrasts within and between species, and outline where additional information is needed to fully exploit the lens-based isotope archive.

## Structure and formation of the vertebrate eye lens

The cellular structure of the vertebrate eye lens is conserved across taxa, despite evolving to suit numerous ecological niches (*Blundell et al., 1981*). Within the eye, lenses are cellular organs that serve as a focusing device, allowing images to form on the retina. In order to fulfill this role, lenses must be transparent and have a high refractive index (*Bloemendal et al., 2004*; *Kröger, 2013*). Growth of the lens begins during early embryonic development (*Grainger, 1992*) and stems from the ectoderm that overlies the optic cup, which subsequently inverts and pinches off to form a hollow vesicle (*Bassnett & Beebe, 1992*; *Bloemendal et al., 2004*). Cells occupying the posterior region of the lens elongate to form primary lens fiber cells, which subsequently fill the vesicle (*Bloemendal et al., 2004*). The anterior region of the lens comprises a monolayer of epithelial cells which divide in a region located anterior to the equator (*Bloemendal et al., 2004*). Newly formed cells

elongate forming secondary fiber cells, which overlay the primary fibers (*Bron et al., 2000*; *Kröger, 2013*). Cells in the same layer connect at the poles of the lens in an area referred to as the suture plane. Suture planes result in some light scattering even in healthy lenses, and are believed to serve as pathways of extracellular transport (*Vaghefi et al., 2012*).

Fiber cells are densely packed and arranged in concentric layers with older cells occupying the central core region, and younger cells sequentially added on top (*Bloemendal et al., 2004*; *Kröger, 2013*). This process continues throughout the lifetime of all vertebrates, including humans (*Augusteyn, 2010*), resulting in a lifetime increases in tissue volume (*Bassnett & Beebe, 1992*), although critically, lens growth rate reduces throughout ontogeny (*Bron et al., 2000*), potentially producing near isometric growth of lens diameter compared to total body size. During fiber cell differentiation, protein-coding genes are expressed in a regular manner resulting in variable relative proportions of proteins between the core/nucleus and outer cortex (*Bloemendal et al., 2004*). These soluble proteins are collectively referred to as the crystallins. During the last phase of fiber cell differentiation, degradation of the membrane-bound organelles occurs in a process which closely resembles the early stages of apoptosis (*Bassnett, 2002*). Loss of nuclei, mitochondria and ribosomes removes potential light scattering objects, and is necessary for the lens to achieve transparency (*Bassnett & Beebe, 1992*; *Bassnett, 2002*; *Bloemendal et al., 2004*). Primary fiber cells are the first to lose their organelles, resulting in an organelle free zone (OFZ) (*Bassnett, 2002*). As development proceeds, the OFZ also envelops secondary fiber cells (*Bassnett, 2002*). In fishes, the depth horizon of fiber cell denucleation is approximately 90% of the lens radius (*Schartau et al., 2009*). Beyond the OFZ, in the outer c.10% of the lens radius, the process of cell denucleation is incomplete. After the loss of organelles, fiber cells can no longer synthesize or degrade proteins, and crystalline proteins therefore persist throughout the organism's lifetime (*Bloemendal et al., 2004*).

## Lens proteins

Lens fiber cells have an unusually high cytosolic protein content which, in fish, varies from approximately 0.13 g ml$^{-1}$ at the outer edge to 1.05 g ml$^{-1}$ at the core (*Bloemendal et al., 2004*). All vertebrate lenses contain $\alpha$, $\beta$ and $\gamma$ crystalline proteins, commonly known to as the classic or ubiquitous crystallins (*Lynnerup et al., 2008*; *Bloemendal et al., 2004*). Other crystalline proteins exist within the lens, however these are taxon-specific (*Bloemendal et al., 2004*). The increasing refractive gradient from lens edge to core is associated with the water content and protein concentration of the fiber cells. Water content is controlled by regulation and post-translational modification of aquaporin-0 (*Ball et al., 2004*; *Hedfalk et al., 2006*; *Törnroth-Horsefield et al., 2010*; *Gutierrez, Garland & Schey, 2011*). The core region of the lens has a high protein and low water content, and is often hard and incompressible (*Fernald, 1985*; *Bloemendal et al., 2004*). Once the proteins have obtained this high concentration, they are generally not re-soluble (*Bloemendal et al., 2004*) and the nucleus achieves a uniform, relatively high, refractive index (*Fernald, 1985*). Moving away from the core, cytosolic protein concentration decreases (*Philipson, 1969*), and water content increases, resulting in the reduction in the refractive index. Close to the lens surface, cytosolic protein concentration decreases rapidly (*Kröger, 2013*). This results

in a steep refractive gradient between the outer edge of the core region, and the outer cortex (*Fernald, 1985*). This gradient is essential in order to maintain visual functioning in fishes as the cornea provides no refractive power in water (*De Busserolles et al., 2013*). Consequently in fully aquatic animals the lens, with its variable refractive gradient from core to edge, is solely responsible for light refraction, and is generally spherical or sub-spherical in fishes (*Nicol, 1989*). Proteins must therefore exist both in a concentrated solid state in the core, and a gelatinous solution at the outer cortex (*Lynnerup et al., 2008*), producing a tissue consisting of two physically distinct zones (*Fernald, 1985*). The shape of the outer cortex is maintained by the lens capsule (*Fernald, 1985*), whereas the central core region is a dense, stable structure. In the eye, the epithelial basement membrane completely encloses the lens, resulting in the absence of blood vessels (*Lynnerup et al., 2008*). Desquamation of ageing cells is therefore restricted as no metabolites are transported to the lens, preventing cell degradation (*Beebe, 2003*). Thus, the protein-filled lens fiber cells are not remodeled throughout ontogeny. The eye lens is thus a sequentially formed, metabolically inert tissue, almost pure protein, and suitable for hosting ontogenetic biochemical information.

The potential for deriving life history information from the inert, organic, and incrementally formed fish eye lens has been explored, albeit less intensively than for other incrementally grown tissues (*Tzadik et al., 2017*). The trace element composition of eye lenses has been studied in the context of fish migration and stock discrimination studies (*Dove & Kingsford, 1998*; *Kingsford & Gillanders, 2000*; *Gillanders, 2001*). Differences in absolute and relative trace element compositions between lens, otolith and other biomineralised tissues were found, as expected based on the fundamental chemical differences between host matrices (proteins compared to mineral lattices). Lenses are exposed to the aqueous humor implying potential for post-depositional exchange of weakly-bonded ions, particularly considering the relatively permeable nature of lens proteins (*Dove & Kingsford, 1998*; *Tzadik et al., 2017*). While lenses may be a less suitable host than otoliths for the more ionically-bonded metal ions such as Sr, Mg and Ba which form the bulk of otolith microchemical work, covalently bonded ions such as Pb and Hg maybe more strongly bonded to proteins, and thus more effectively recorded in lens tissues. The trace element composition of eye lenses has received relatively little attention despite calls to consider lenses as suitable targets, particularly for covalently bound metal ions (*Dove & Kingsford, 1998*; *Tzadik et al., 2017*).

The stable isotope composition of carbon and nitrogen in lens proteins has been explored in an attempt to reconstruct feeding chronologies in squid (*Parry, 2003*; *Hunsicker et al., 2010*; *Onthank, 2013*). *Parry (2003)* identified an increasing trend in $\delta^{15}N$ values with body size in sequential samples of lens proteins from *Ommastrephes bartramii* and *Sthenoteuthis oualaniensis*, suggesting that ontogenetic trophic shifts are reflected and preserved within eye lens chemistry. Whilst *Onthank (2013)* identified no common ontogenetic trend in stable isotope composition among squid, individual specimens did exhibit ontogenetic variations in the stable nitrogen isotope composition of lens proteins, suggesting changes in diet over the individual's lifetime. *Hunsicker et al. (2010)* applied similar methodologies to study the commander squid (*Berryteuthis magister*) in the eastern Bering Sea, identifying an increase of ~1 trophic position between juvenile and adult life stages. A similar approach

was adopted to demonstrate recovery of whole-life stable isotope records from teleost fishes (*Wallace, Hollander & Peebles, 2014*). Red grouper (*Epinephelus morio*), gag (*Mycteroperca microlepis*), red snapper (*Lutjanus campechanus*), and white grunt (*Haemulon plumierii*) lenses were examined, showing consistency between left and right eye lenses, and recovering increasing $\delta^{15}$N values from lens nucleus (core) to cortex, as expected given anticipated increases in trophic position with size or age. The inert nature of lens core proteins formed in the early juvenile period also opens the potential for radiocarbon-based age estimation. *Lynnerup et al. (2008)* and *Kjeldsen et al. (2010)* both reported successful application of bomb radiocarbon dating to human eye lenses. More recently, *Nielsen et al. (2016)* used radiocarbon dating on the eye lens nuclei of 28 female Greenland sharks (*Somniosus microcephalus*), estimating longevity to be on the order of hundreds of years.

Elasmobranchs do not possess large, calcified otoliths suitable for age determination and ontogenetic investigation (*Cailliet & Goldman, 2004*), consequently life histories of elasmobranchs are generally much more poorly understood than those of teleosts. Elasmobranch vertebrae and fin spines grow incrementally, and can offer sclerochronological detail, providing most of the known methods to age elasmobranchs (*Cailliet & Goldman, 2004*). However many elasmobranchs do not have fin spines, and poorly mineralized vertebrae present challenges for accurate and precise age determination. Vertebrae do offer a medium for retrospective isotopic analysis (e.g., *Kim et al., 2012*), but many species have small vertebrae, and the mass requirements of analysis coupled with the difficulty of relating vertebral position to body size or age, and the need to remove carbonated bioapatite from mineralized vertebral samples limits the number of samples available for ontogenetic study, ultimately reducing temporal resolution. Functioning eyes are generally essential for hunting in most fishes, and there is selective pressure to develop eyes early in ontogeny. Eye lenses may therefore be particularly well suited to the reconstruction of early life history ecology in fishes. In many elasmobranchs, the yolk or placenta provides nourishment to the developing pup prior to emergence from the egg case (oviparous) or birth (viviparous/aplacental viviparous) (*Hamlett et al., 1993*). Nutrient assimilation driving embryonic development in elasmobranchs is thus characterized principally by the chemical properties of the prey of mature females during gestation or the period of yolk sac formation. The chemical composition of core regions of incrementally grown tissues of elasmobranchs therefore reflects maternal trophic geography during egg formation. The isotopic record at the core of the eye lens is particularly interesting as mature female elasmobranchs are often elusive, and therefore difficult to observe and study using traditional techniques.

The short review of eye lens structure and development suggests that lens proteins are promising hosts for the recovery of stable isotope life histories of marine animals, particularly for juvenile life stages. However, several additional factors currently limit the use of lens protein as a useful target for the recovery of life history stable isotope records:

(1) As lenses do not contain regular growth increments, age cannot be inferred directly from the lens alone. Body length may potentially be inferred from lens diameter if the relationship between lens diameter and body size is known. Body length then may be related to age based on age:size relationships.

(2) Lens structure and specifically the hydration of lenses may vary between taxa. Sample preparation and handling protocols may therefore need to reflect changes in lens composition and structure across taxa.

(3) Lens proteins are composed of different proportional combinations of amino acids to other commonly sampled tissues which could result in systematic offsets in isotopic composition. The isotopic offset between lens proteins and muscle (or other) tissues should be determined.

(4) To improve confidence in a relatively new isotopic host tissue, recovery of expected patterns of isotopic variance within and among individuals and species should be demonstrated.

In the remainder of this study we address the points above using lens tissues from elasmobranch and teleost fishes.

## MATERIALS AND METHODS

Samples for this study were obtained opportunistically from research cruises and samples available though other projects. Deep-water fishes often possess relatively large eyes for a given body size, and consequently are attractive subjects for exploratory analyses of eye lens chemistry. Lens and body measurements were obtained from two deep-water teleost species, the black scabbardfish (*Aphanopus carbo*) and the roundnose grenadier (*C. rupestris*), and two shallow water elasmobranch species, the porbeagle shark (*Lamna nasus*) and spurdog (*Squalus acanthias*). *A. carbo* ($n = 19$) and *C. rupestris* ($n = 29$) lenses were sampled from fish captured from deep-water trawl surveys off the west Scotland continental slope in 2012 and 2013 onboard the MRV Scotia (survey methods reported in *Neat et al., 2010*). Body measurements were recorded, and a randomly-chosen sub-set of lenses from both species was processed for stable isotope analysis ($n = 8$ and $n = 11$, for *A. carbo* and *C. rupestris*, respectively).

*L. nasus* ($n = 30$) lenses and associated body size data were collected from sharks incidentally by-caught by off-shore commercial gill-net fisheries within the Celtic Sea between 2011 and 2014, landed under dispensation in association with dedicated Cefas-led scientific fishery studies (*Bendall et al., 2012*; *Ellis et al., 2016*). Lenses and allometric data were recovered from *S. acanthias* ($n = 101$) incidentally by-caught aboard inshore commercial long line fisheries in the southern North Sea in 2013. A subset of 19 *S. acanthias* individuals were randomly selected for isotopic investigation and white muscle and lens samples extracted. A proportion of by-caught *S. acanthias* were pregnant females, providing an opportunity to compare white muscle and lens tissues in a further 19 developing embryos.

To compare lens morphology among species we obtained lenses from small numbers of individuals from a range of taxa. Lenses were extracted from cod (*Gadus morhua*, $n = 2$), haddock (*Melanogrammus aeglefinus*, $n = 4$), hake (*Merluccius merluccius*, $n = 4$) and conger eels (*Conger conger*, $n = 2$) obtained from commercial fishermen operating in the English Channel and North Atlantic. Monkfish (*Lophius americanus*) lenses ($n = 6$) were provided by The School for Marine Science and Technology (SMAST), at the University
of Massachusetts (Dartmouth). Oceanic white-tip shark (*Carcharhinus longimanus*) lenses were obtained from one by-caught specimen caught off Wallis and Futuna in the South Pacific, in 2012, whilst blue shark (*Prionace glauca*) samples ($n = 2$) were obtained from individuals caught by Spanish and Portuguese long-lining vessels operating off the Canary Islands in 2014.

We recognize that sample sizes for morphological comparisons are extremely small, but the aim was to compare lens structure among a range of fish taxa. We do not anticipate that gross lens morphology and hydration is highly variable between individuals (and experience of sampling multiple individuals of the same species for isotopic analysis confirms this) and so we believe that the data collected provides a good indication of variability in lens structure and permeability among the sampled taxa.

## Tissue preparation

An incision was made along the length of the cornea of each fish, allowing the lens to be removed using forceps. To test allometric relationships, the maximum diameter of the lens was measured to the nearest 0.25 mm with calipers. Once excised, the lenses were frozen at $-80\,°C$ overnight, before being freeze-dried for 12 h. Once freeze-dried, lenses were embedded in epoxy resin (Epofix; Electron Microscopy Sciences, Hatfield, PA, USA) to secure the lenses for sectioning.

To compare lens core permeability between species, eye lenses extracted from a range of teleost and elasmobranch species were freeze-dried and embedded in blue dyed Epoxy resin. Penetration of the blue dye provides a relative measure of lens permeability to a viscous, hydroscopic fluid after lyophilisation.

Embedded lenses were sectioned using two diamond wafering blades (0.3 mm thick) mounted upon a low-speed precision saw (Isomet; Buehler, Lake Bluff, IL, USA), separated by a 2 mm spacer. The double blade system was used to produce a thin section, with one of the blades running through the absolute core of the lens. This produced a section revealing the lens structure. Dye-embedded lenses were also sectioned using the same instrument and settings, and the resultant thin section imaged using a Panasonic Lumix FZ200 Bridge Camera.

Lenses were sampled for sequential analyses from seven *A. carbo*, 11 *L. nasus* and 11 *C. rupestris* individuals. In preparation for stable isotope analysis, a second section was made across the surface of the exposed lens produced by the first double-blade cut, producing a section of lens, approximately 2 mm thick and 2 mm wide with the lens core at its center. The lens strip was then sequentially sectioned by removing slivers of lens material, measuring approximately 0.5 mm in width, using a scalpel. Sub-sampled segments of dried lens tissue were weighed into tin capsules for stable isotope analysis. Samples of pure resin were also submitted for analysis to monitor potential contamination.

For *S. acanthias*, white muscle and lens samples were taken from 19 embryonic sharks. Muscle samples were rinsed ten times in deionized water to remove free ammonia, and centrifuged for 10 min, before being rinsed a further ten times. Lens samples were removed from the lens by physical peeling with no resin embedding. The muscle and lens samples were frozen, lyophilized, and weighed prior to analysis.

## Analytical methods

Stable carbon and nitrogen isotope ratios of teleost and *L. nasus* lens samples were determined at OEA Laboratories Ltd. International reference material glutamic acid (USGS40; *Qi et al., 2003*) were analyzed for every ~10 unknown samples in each analytical sequence. Isotope ratios are expressed in δ notation (‰), relative to the international standards of V-Pee Dee belemnite and air for carbon and nitrogen, respectively. Experimental precision for *C. rupestris* and *L. nasus* sample analysis was 0.21‰ for carbon and 0.27‰ for nitrogen. Experimental precision for *A. carbo* sample analysis was 0.12‰ and 0.16‰ for carbon and nitrogen, respectively (standard deviation of replicate USGS40 standards).

All *S. acanthias* lens and embryo muscle samples were analyzed at the NERC Life Sciences Mass Spectrometry Facility (LSMSF) in East Kilbride, using a Pyrocube Elemental Analyzer (2013, Elementar, Hanau, Germany) and Delta XP (Thermo Electron) mass spectrometer (2003). Drift correction using a 3-point normalization was performed using three in-house standards (GEL, ALAGEL and GLYGEL) encompassing a range of isotope compositions, run every ten samples, and a suite of GELs of different sizes were used to correct samples for linearity (*Werner & Brand, 2001*). GEL is a gelatin solution, ALAGEL an alanine-gelatine solution, and GLYGEL a glycine-gelatine solution. Four differently sized USGS 40 glutamic acid standards (*Qi et al., 2003*; *Coplen et al., 2006*) were used as independent checks of accuracy and to acquire the calibration for N and C content. All data are reported with respect to the international standard of AIR for $\delta^{15}N$ and V-PDB for $\delta^{13}C$. Results are reported in δ notation (*McKinney et al., 1950*). Experimental precision was 0.14‰ for carbon and 0.13‰ for nitrogen (standard deviation of laboratory standard replicates).

To account for inter-laboratory variability, replicate glutamic acid standards from The University of Southampton were analyzed at both facilities. Mean $\delta^{15}N$ and $\delta^{13}C$ reported for standards run at OEA Laboratories Ltd was −3.99‰ (±0.23) and −12.99 (±0.27) respectively, while the same standards run at the LSMSF facility returned mean $\delta^{15}N$ and $\delta^{13}C$ values of −3.84‰ (±0.16) and −13.03 (±0.24), respectively. Therefore no inter-laboratory corrections were applied.

## RESULTS

### Allometric growth

The relationship between total length (TL, mm) and lens diameter (LD, mm) was linear across the size range sampled (Fig. 1). The best fit linear relationships between body length and lens diameter are:

$$\text{\textit{Aphonopus carbo}} \quad LD = -3.713 \, (\pm 1.279) + 0.0185 \, (\pm 0.013) * TL \tag{1}$$

$$\text{\textit{C. rupestris}} \quad LD = -0.369 \, (\pm 0.142) + 0.0108 \, (\pm 0.003) * TL \tag{2}$$

$$\text{\textit{L. nasus}} \quad LD = 4.461 \, (\pm 0.733) + 0.0070 \, (\pm 0.004) * TL \tag{3}$$

$$\text{\textit{S. acanthias}} \quad LD = 0.479 \, (\pm 0.181) + 0.0109 \, (\pm 0.002) * TL. \tag{4}$$

In the case of *C. rupestris* and both shark species, a near-full size range of individuals was sampled. *C. rupestris* specimens as small as 50 mm were obtained, representing the
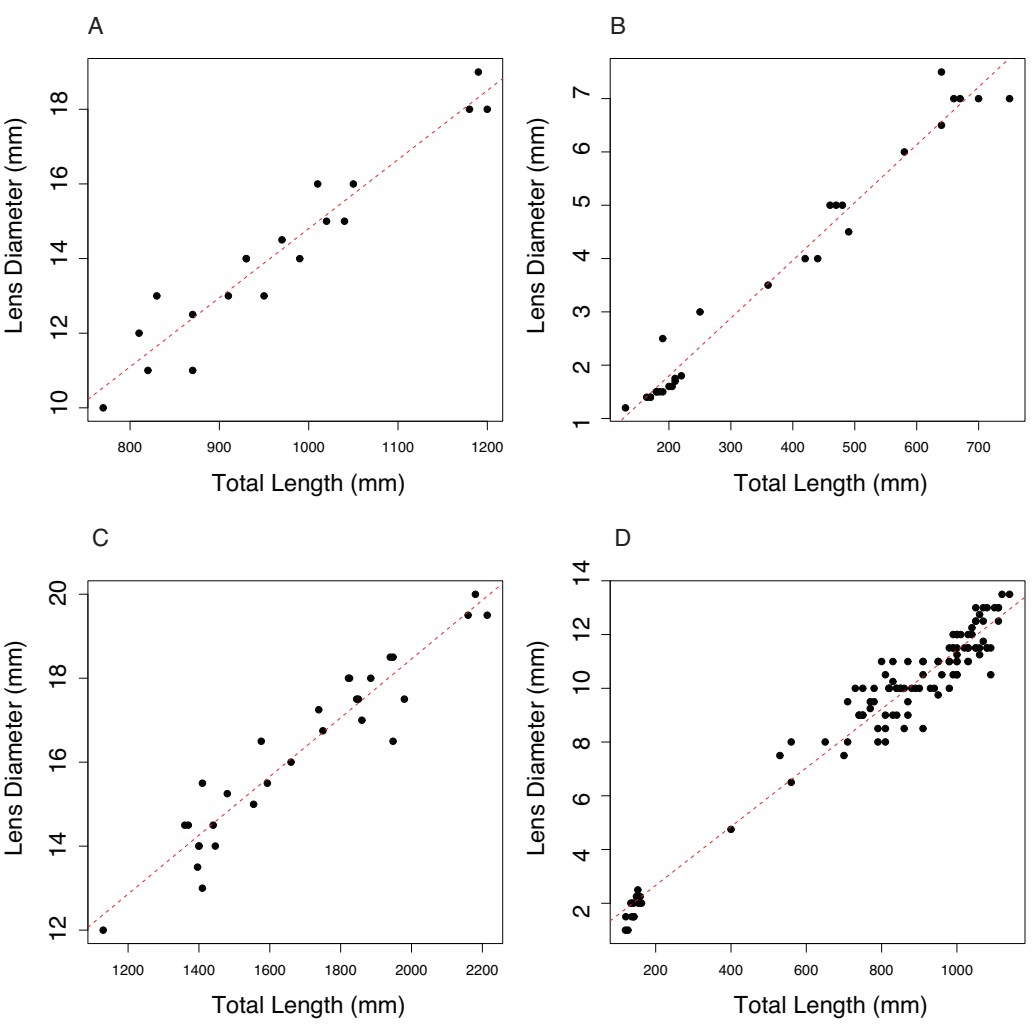

**Figure 1** **The relationship between total length (TL) and lens diameter (LD) in four fishes.** The relationship between total length (TL) and lens diameter (LD) for (A) the black scabbard fish, *A. carbo*, ($n = 19$, $r^2 = 0.917$, $p = < 0.001$). (B) The roundnose grenadier, *C. rupestris*, ($n = 29$, $r^2 = 0.973$, $p \leq 0.001$). (C) The porbeagle shark, *L. nasus* ($n = 30$, $r^2 = 0.901$, $p \leq 0.001$). (DT) The spurdog, *S. acanthias* ($n = 101$, $r^2 = 0.959$, $p \leq 0.001$).

early juvenile life history stage of this species. Similarly, a juvenile *L. nasus* measuring 1,130 mm was captured (size at birth is estimated at ~800–900 mm (*Ebert, Fowler & Compagno, 2013*)). We also sampled developing *S. acanthias* embryos, thus we are confident that the allometric growth equations (Eqs. (2)–(4)) accurately reflect the relationship between lens diameter and body size for *S. acanthias*. An analysis of covariance (ANCOVA) revealed that lens diameter is influenced by individual size (TL) and species ($F_{1,189} = 9275.18$, $p \leq 0.001$ and $F_{3,189} = 264$, $P \leq 0.001$, respectively) as the interaction of these variables ($F_{3,189} = 38.79$, $p \leq 0.001$). Thus, the slopes and intercepts of Eqs. (1)–(4) differ from one another, and the allometric relationship between body size and lens diameter differs between species.

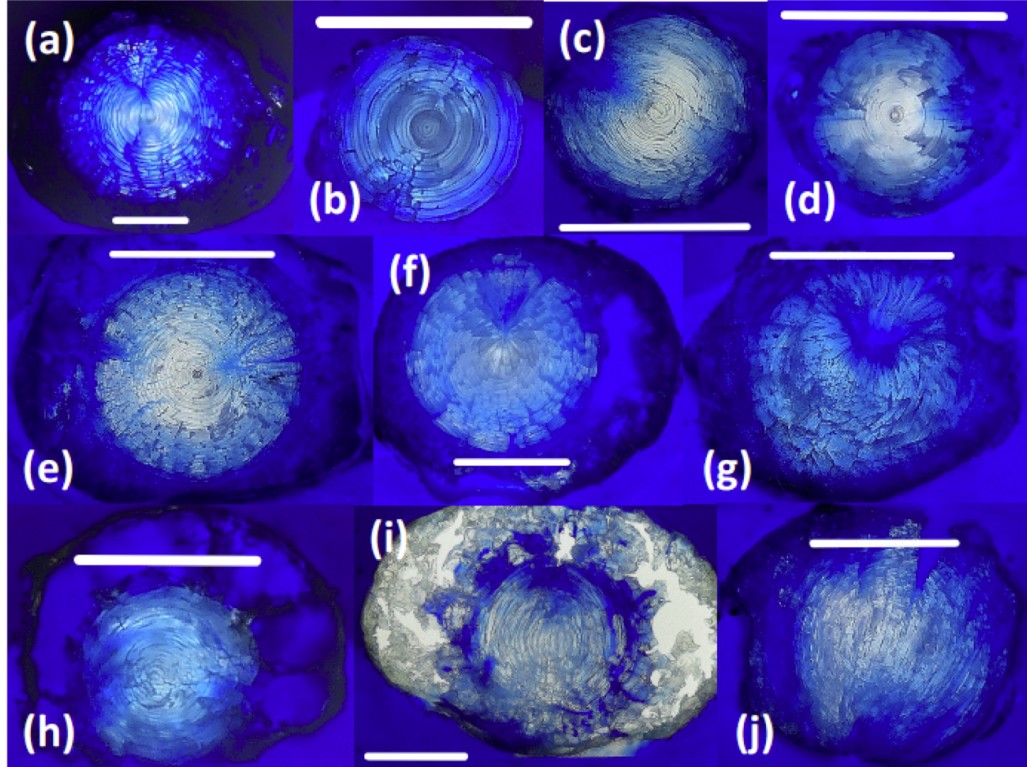

**Figure 2** **Thin sections of teleost and elasmobranch lenses embedded in pigmented Epoxy resin.** (A) Black scabbardfish (*Aphanopus carbo*). (B) Monkfish (*Lophius americanus*). (C) Cod (*Gadus morhua*). (D) Conger eel (*Conger conger*). (E) Hake (*Merluccius merluccius*). (F) Haddock (*Melanogrammus aeglefinus*). (G) Porbeagle (*Lamna nasus*). (H) Spurdog (*Squalus acanthias*). (I) Blue shark (*Prionace glauca*). (J) Oceanic white-tip shark (*Carcharhinus longimanus*). White lines are 5 mm scale bars.

## Comparison of lens morphology and permeability

Sectioned teleost lenses reveal well-defined, concentric banding, which is enhanced during freeze-drying as weaknesses between fiber cell layers promote concentric fractures (Fig. 2). Assuming that within-species variation in lens structure is minimal, distinct differences in lens morphology and the structural response of the lens to freeze-drying are seen between species: *G. morhua*, *C. conger*, and *L. americanus* lenses all have clear, concentric bands throughout, radiating from the lens core to edge (outer cortex). *M. merluccius*, *M. aeglefinus*, and *A. carbo* lenses also display clear concentric banding throughout the solid core region of the lens, but with no clear structure within the hydrated outer cortex region. *M. merluccius*, *M. aeglefinus*, and *A. carbo* lenses were relatively hydrated compared to *G. morhua* and *C. conger* lenses, and required careful handling to avoid puncturing the lens membrane, which would result in the loss of the gelatinous material.

All elasmobranch lenses included in this comparison presented difficulties associated with handling prior to freeze-drying, due to the large proportion of gelatinous material.

The core region of all elasmobranch lenses examined here was generally less dense than that of the teleosts. Sampled elasmobranch lenses also show less concentric banding, and increased hydration relative to teleost lenses (Fig. 2). The relative penetration of dyed resin into the lens differs according to the species examined. For example, the lenses of *G. morhua*, *C. conger*, and *M. merluccius* are relatively impermeable, (Fig. 2). Conversely, in lenses of *L. nasus*, *C. longimanus*, *M. aeglefinus*, and *L. americanus*, resin penetrated to the core of the lens, possibly via fractures produced during freeze-drying. A qualitative ranking of susceptibility to resin contamination was (most susceptible first): *P. glauca* > *L. nasus* > *L. americanus* > *M. aeglefinus* > *S. acanthias* > *A. carbo* > *C. longimanus* > *M. merlucuius* > *C. conger* > *G. morhua*. We infer that permeability to resin reflects either increased fracturing of more hydrated lenses during freeze drying and/or variations in lens porosity.

## Stable isotope data

Penetration of stained resin into the lens (Fig. 2) indicates the potential for contamination of lens protein with carbon and nitrogen derived from the resin. The isotopic composition of pure resin differs markedly from that of lens protein, allowing us to use a simple mass balance approach to estimate the relative proportion of resin in each sub-sample:

$$\delta^{13}C_s = A * \delta^{13}C_r + B * \delta^{13}C_l \tag{5}$$

where $\delta^{13}C_s$ represents the isotopic composition of the potentially mixed sample (i.e., lens and resin), $\delta^{13}C_r$ represents the isotopic composition of carbon in the pure resin ($-29.5‰$) and $\delta^{13}C_l$ represents the isotopic composition of carbon in a pure lens sample (estimated from core values for each species), and $A$ and $B$ represent the relative proportion of resin and lens in each sample, respectively. As Eq. (5) represents a two-component mass balance, it can be solved explicitly to yield the proportional contributions of resin-derived carbon in any mixed sample given isotopic compositions for $\delta^{13}C_s$, $\delta^{13}C_r$ and $\delta^{13}C_l$. The equation above assumes that core and edge $\delta^{13}C$ values do not differ significantly compared to the isotopic difference between resin and lens protein, this is most likely met where the isotopic composition of the resin is relatively low (as here), but the relative $\delta^{13}C$ values between lens and resin should be noted prior to any assessment of contamination. We cannot apply similar mass balance approaches with $\delta^{15}N$ values, as we expect core and edge $\delta^{15}N$ values to vary systematically (but to an uncertain degree) with body size and trophic level.

Isotope data suggest that resin contamination occurred and penetrated through to the core in all lenses (Table 1). If the proportion of resin estimated in a given sample exceeded 20%, that sample was excluded from any further analyses to avoid excessive errors associated with large corrections. Where the percentage of resin within a particular sample was less than this threshold (<20%), the degree of contamination was corrected for using the following equation:

$$\delta^{13}C_c = \frac{(\delta^{13}C_s * CN_s) - (\delta^{13}C_r * (CN_s - 3.7))}{3.7} \tag{6}$$

 where $\delta^{13}C_c$ represents the corrected isotopic composition of a given sample, and subscript "s" and "r" denote uncorrected carbon isotope composition of the sample, and pure resin,

**Table 1** **Effect of resin contamination on stable isotope compositions of sectioned eye lens proteins.**
Comparison of mean lens core $\delta^{15}N$ and corrected $\delta^{13}C$ values (and associated standard deviations) for
*A. carbo*, *C. rupestris* and *L. nasus* (proportion resin <20%). The stable isotope composition of pure epoxy
resin is also displayed. As a measure of resin contamination of core samples, mean proportion resin esti-
mates are presented (calculated using the mass balance approach).

| | Pure resin | A. carbo (n = 11) | | C. rupestris (n = 11) | | L. nasus (n = 6) | |
|---|---|---|---|---|---|---|---|
| | | Mean | SD | Mean | SD | Mean | SD |
| $\delta^{15}N$ | −2.28 | 9.98 | 1.44 | 6.83 | 1.57 | 12.74 | 1.13 |
| $\delta^{13}C$ (corrected) | −29.49 | −20.70 | 0.72 | −19.17 | 0.84 | −18.14 | 0.77 |
| Prop. Resin | NA | 0.12 | 0.07 | 0.12 | 0.08 | 0.13 | 0.26 |

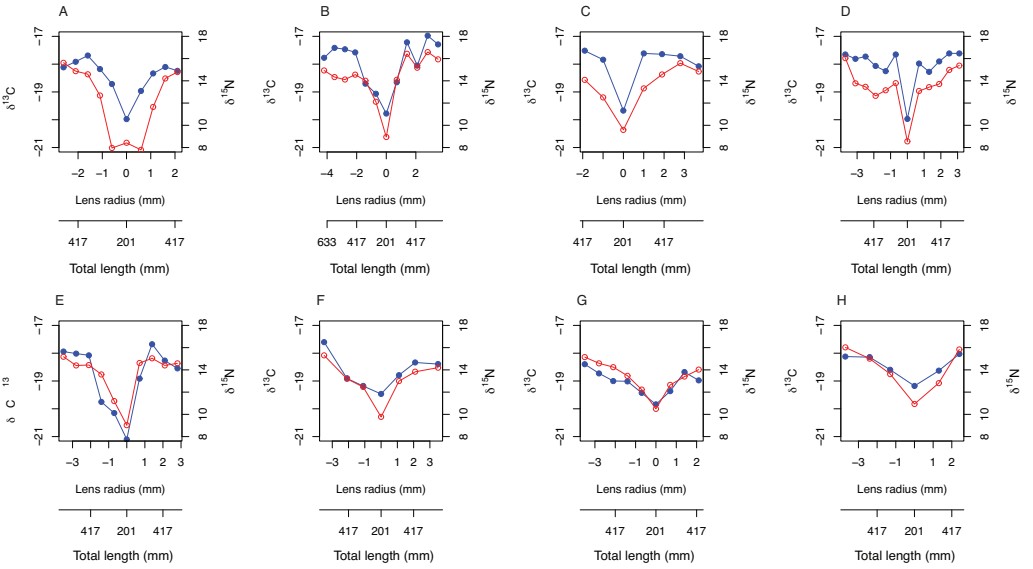

**Figure 3** **Bivariate plots (A–H) of $\delta^{15}N$ (blue) and $\delta^{13}C$ (red) variability through *A. carbo* lenses.** The
corresponding estimate of individual fish total length at time of tissue formation was recovered from rela-
tionship shown in Fig. 1A, and the location of the sample from the lens transect.

respectively. 3.7 reflects the C:N ratio of a pure protein. Lenses of *C. rupestris* were most
susceptible to resin contamination with 39% of all samples containing >20% resin,
followed by *L. nasus* with 36% of samples exceeding 20% contamination. Relatively little
resin contamination was observed in *A. carbo* lenses, and only 13% of these samples were
omitted from the dataset due to contamination.

## Ontogenetic trends in stable isotope compositions across lenses
Symmetrical ontogenetic trends in resin-corrected $\delta^{15}N$ and resin-corrected $\delta^{13}C$ values
centered around the lens core were recovered for all sampled *A. carbo* lenses. The total
length of the fish associated with each lens sub-sample was estimated using Eq. (1) and the
corresponding distance from lens core (mm). In general, $\delta^{15}N$ and $\delta^{13}C$ values increase
with body size in *A. carbo* with a sharp isotopic excursion associated with the lens core
(Fig. 3).

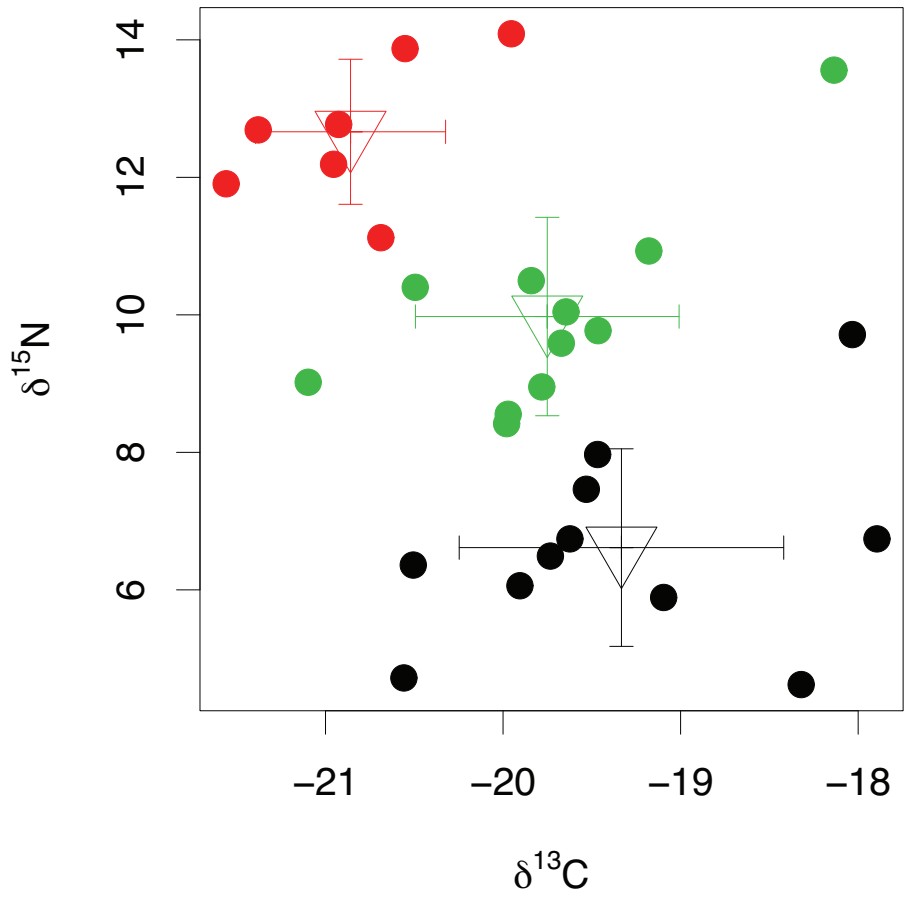

**Figure 4** **Stable isotope compositions of lens tissues between species.** Comparison of lens core $\delta^{13}C$ and $\delta^{15}N$ values for *C. rupestris*, *A. carbo* and *L. nasus*. Error bars show standard deviation.

## Among-species comparisons of lens isotope compositions

Among-species comparisons in the isotopic composition of eye lens proteins were carried out between *C. rupestris*, *L. nasus* and *A. carbo*. Due to the high resin content of sequential lens samples of *C. rupestris* and *L. nasus*, only isotope values from the lens core, representing early juvenile compositions, are compared among species. The isotopic compositions of the lens core of the three sampled species are distinct (Fig. 4) (one-way ANOVA, $F = 8.01$, $df = 2$, $p = 0.001$, and $F = 44.13$, $df = 2$, $p \leq 0.001$, for carbon and nitrogen, respectively). Lowest mean nitrogen isotope ratios are seen in *C. rupestris* (mean = 7.95‰ ± 1.68), followed by *A. carbo* (mean = 11.58‰ ± 1.23) and *L. nasus* (mean = 15.22‰ ± 1.81). Lens core carbon isotope compositions for *C. rupestris* and *L. nasus* are distinct from one another, however there is considerable isotopic overlap between *A. carbo*, and both *C. rupestris* and *L. nasus*.

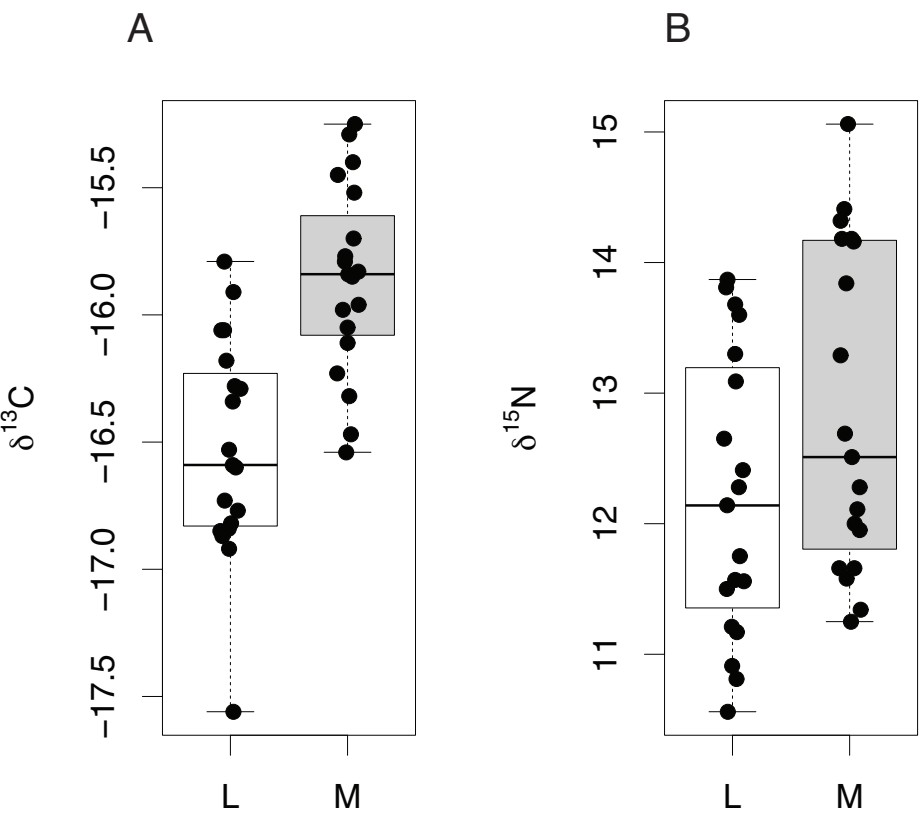

**Figure 5 Tissue-specific differences in isotopic compositions in embryonic sharks.** Comparison of lens and muscle $\delta^{13}C$ (A) and $\delta^{15}N$ (B) values from 19 *S. acanthias* embryos.

## Isotopic fractionation between lens and muscle tissue

The isotopic composition of lens and muscle samples from 19 *S. acanthias* embryos was compared. Lens and muscle tissue formed *in utero* are derived from the same nutrient source (egg yolk) and have similar isotopic incorporation times, not exceeding the species' gestation period. Due to the small size of pup lens samples, whole lenses were analyzed. A lipid correction model based on *S. acanthias* (*Reum, 2011*) was applied to the muscle data. The isotopic compositions of lens protein from *S. acanthias* show similar variability in both $\delta^{15}N$ and $\delta^{13}C$ values compared to muscle (Fig. 5, Levene's test of homogeneity of variances $F$ value $= 0.48$ in both cases, $p > 0.45$). Muscle tissue was enriched in both $^{13}C$ and $^{15}N$ relative to lens protein (paired $t$-test, mean offset in carbon $= 0.66 \pm 0.22‰$, $t = 12.55$, degrees of freedom $= 18$, $p \leq 0.001$: mean offset in nitrogen $= 0.66 \pm 0.36\%$ $t = 7.56$, degrees of freedom $= 18$, $p \leq 0.001$).

Finally, $\delta^{15}N$ and $\delta^{13}C$ values from the lens core of mature *C. rupestris* ($n = 11$) were compared with existing measurements of $\delta^{15}N$ and $\delta^{13}C$ values of muscle tissue from early juvenile *C. rupestris* specimens measuring $\leq 50$ mm, recovered from the same region ($n = 5$). Lens tissues displayed a greater range in both $\delta^{15}N$ and $\delta^{13}C$ values compared to muscle (Fig. 6), but variances were not significantly different (Levene's test

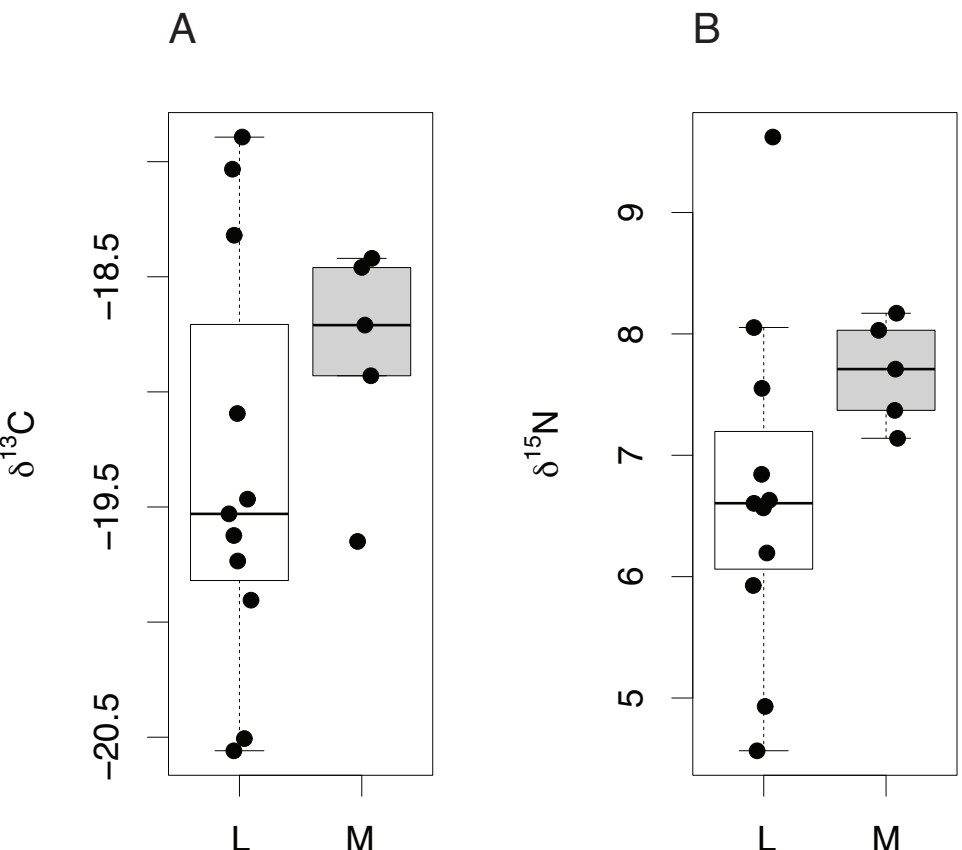

**Figure 6** **Tissue-specific differences in isotopic compositions in larval and adult fish.** Comparison of lens core (L) and muscle (M) δ13C (A) and δ15N (B) values from *C. rupestris* specimens measuring ≤ 50 mm. Eye lenses and muscle samples were obtained from specimens caught during the deep-water survey onboard MRV Scotia; lenses were excised from fish caught in September 2012, and the muscle originates from specimens caught during September 2013.

of homogeneity of variances $F$ value = 1.4 ($\delta^{13}$C) and 1.8 ($\delta^{15}$N), in both cases, $p > 0.15$). Lenses were again generally depleted in $^{13}$C and $^{15}$N, but mean $\delta^{15}$N and $\delta^{13}$C values did not differ significantly between lens and muscle proteins ($\delta^{15}$N − t = 0.495, $p = 0.332$, $\delta^{13}$C − t = −1.001, $p = 0.332$).

## DISCUSSION

We initially identified the following issues that need to be addressed before eye lens tissues can be used as an effective target for ontogenetic stable isotope analyses:

(1) The relationship between lens diameter and body size should be tested across taxa.
(2) The structure of lenses should be compared between taxa, and a consistent sampling methodology should be determined.
(3) The isotopic offset between lens and muscle tissue should be calculated across taxa.

In addition, we suggested that isotopic differences expected between and within individuals should be demonstrated across a range of taxa.

Below we address each of these issues in turn.

## Allometry

Lens accretion occurs in linear proportion to somatic growth in each of the four species studied in detail (*A. carbo*, *C. rupestris*, *L. nasus* and *S. acanthias*). Linear allometry between lens radius and body size indicates that the lens presents a near whole-life chronology of chemical information with greater temporal resolution during periods of rapid body size growth. Allometric growth allows retrospective estimation of the approximate size of the fish corresponding to a particular location within the lens. The slope and intercepts of the lens to body size (and therefore potentially lens diameter to age) relationship differ between species, requiring allometric relationships to be estimated for each new study species, at least until sufficient species have been assessed to develop predictive relationships.

## Lens structure and sampling methodology

Fish eye lenses are relatively difficult tissues to handle due to their partially hydrated, gelatinous exterior, hard internal core and spherical shape. Two alternative sampling processes have been proposed: peeling layers from the lens, developed by *Parry (2003)* and later applied by *Wallace, Hollander & Peebles (2014)*, and embedding and sectioning comparable to more standard sclerochronological sampling methods. The peeling technique represents a simple and attractive method to sample eye lenses, but sequential peeling techniques are difficult to apply where lenses are either highly crystalline or highly gelatinous. Additionally, it is relatively difficult to accurately control the depth of each sequential layer peeled from a lens, making cross-referencing to age models inaccurate. Embedding lenses in a resin medium then sectioning could provide a more reliable sampling technique, but embedding methods bring the tissue into contact with a potential contaminant (the resin embedding medium) and permeability of the lens to the embedding medium is therefore a critical concern. We found high interspecific diversity in permeability to epoxy resin among lenses (Fig. 2), however teleost lenses are generally denser and less hydrated relative to those of elasmobranchs. The extent of contamination as assessed by carbon isotope mass balance was also highly variable between species, ranging from <10% for dense lenses of *A. carbo*, and core regions in other species, to more than 30% in the hydrated outer portions of *L. nasus* and *C. rupestris* lenses. Visual assessment of resin contamination similarly suggested that the susceptibility to contamination corresponds to the degree of hydration of the lens, and freeze-drying appears to enhance resin contamination during embedding, possibly because of the development of fractures that penetrate circumferential lamellae. The degree of hydration ultimately influences the ability to sample lenses accurately and precisely via both peeling and sectioning.

Lens size may be a driving factor of the relative hydration of lenses and, consequently, the suitability of a particular lens to resin embedding and sectioning. For example, blue shark, oceanic whitetip shark, haddock and black scabbardfish lenses are both the largest lenses and the most hydrated (*a priori* observations). Increased porosity or hydration in large lenses may be associated with the demand for lenses to remain transparent and

reduce light refraction. With increasing size comes an increase in the path length that light must travel through the lens, which in turn increases the potential for light scattering. Maximum protein concentration may thus be limited in larger lenses to a lower level than that of smaller lenses (*Kröger, 2013*) in order to reduce light scattering. The structure of a particular lens will ultimately determine the methodology used, and each species must be addressed on a case-by-case basis. The "peeling" technique (*Parry, 2003*; *Wallace, Hollander & Peebles, 2014*) represents a high-resolution, contamination-free technique, and should be the first choice method. Where "peeling" is not possible, or where accurately resolved distances across the lens are required, the resin-embedding method provides an alternative approach, but only for relatively dense lenses, and the potential for contamination must be assessed. In the case of highly gelatinous lenses, neither methodology discussed in this paper is suitable, at least for recovering sequential samples across the lens.

## Potential for isotopic offset between lens and muscle protein

Estimates of isotopic fractionation between tissues are generally performed across tissues in the same individual. This methodology is relatively difficult for eye lens proteins as (a) integration times for muscle and lens protein are unknown and (b) outer portions of lenses are hydrated, gelatinous and difficult to sample accurately. We therefore took two complementary approaches: comparisons of lens and muscle tissues within developing embryos, and comparing lens and muscle between different individuals but at similar life stages.

Tissues of oviparous embryonic sharks form over relatively similar timescales, are sustained from a single, isotopically constant yolk sac nutrient source, and have no isotopic memory associated with tissue turnover. Embryonic oviparous sharks therefore provide a uniquely constrained opportunity to determine potential isotopic offset effects associated with tissue compositions. Within individual embryos of *S. acanthias*, lens protein was significantly lower in both $\delta^{15}N$ and $\delta^{13}C$ values compared to muscle protein (mean difference $= 0.66 \pm 0.36‰$, $p \leq 0.01$, and $0.66 \pm 0.23‰$, $p \leq 0.01$, for nitrogen and carbon, respectively).

For teleosts, we compared lens protein values in the core regions from adult fish with those of juvenile fish of equivalent sizes (calculated from allometric relationships); we therefore compare across individuals from different year classes, but of equivalent size (Fig. 6). Mean $\delta^{13}C$ and $\delta^{15}N$ values in eye lens proteins were again lower relative to muscle tissue from *Coryphaenoides rupestris* specimens, however the absolute offset is small and effectively obscured by the additional variance associated with un-matched individuals. Consequently, lens protein appears to be lower in both nitrogen and carbon isotope values compared to muscle protein for both elasmobranchs and teleosts, but the isotopic differences are less than $0.75‰$ and $0.25‰$ for $\delta^{15}N$ and $\delta^{13}C$ respectively, and likely to be obscured by between-individual variation and analytical error.

## Recovery of expected isotopic differences within and between species

Transects of $\delta^{15}N$ and $\delta^{13}C$ values across whole lenses of *A. carbo* show a high degree of bilateral symmetry, i.e., different locations of roughly equal radial distance from the

core record similar isotopic compositions. All *A. carbo* specimens analyzed in this study show progressive enrichment in $\delta^{15}$N values consistent with *a priori* expectations of increasing trophic level with body size in this species (*Trueman et al., 2014*), and agree with published trends in the lens nitrogen isotope values of teleosts (*Wallace, Hollander & Peebles, 2014*). The one exception is the presence of elevated core $\delta^{15}$N value in one *A. carbo* specimen (M113, Fig. 2A). We attribute this to the presence of a maternal signal at the absolute core, which may elevate the $\delta^{15}$N value of this sample relative to adjacent tissue. Similar anomalous chemical compositions associated with maternal contributions or larval developmental physiology have been noted in otolith core regions (e.g., *Ben-Tzvi et al., 2007*).

$\delta^{15}$N and $\delta^{13}$C ratios in the lens core differed significantly between the three sampled species (Fig. 4). $\delta^{15}$N values increase from the zooplanktivorous *C. rupestris* to the piscivorous *A. carbo* and are highest in the viviparous *L. nasus*, consistent with expected relative trophic levels for early juveniles of these species. *A. carbo* and *C. rupestris* specimens were both sampled from the north east Atlantic continental slope west of Scotland, but genetic and microchemical data indicate that *C. rupestris* is a lifelong resident species, while *A. carbo* individuals spend their earliest life stages in southern waters around Madeira (*Longmore et al., 2014*). $\delta^{13}$C values successfully distinguish the resident *C. rupestris* from the migratory *A. carbo*. Between-species variation in the isotopic composition of lens core protein is thus greater than among-individual variation, as expected for these functionally distinct species.

## Additional complexities associated with lens-based methodologies

The data presented above, together with previous published studies, clearly demonstrate that eye lenses of teleosts and elasmobranchs can provide retrospective, whole life records of isotopic compositions, which can be compared to isotopic data recovered from alternative tissues. These life history records can be exploited to study spatial and trophic ecology, particularly relating to early juvenile life stages.

The application of lens-based methodologies to investigate chondrichthyan ecology is additionally complicated by the varied reproductive strategies adopted by members of this class. Sharks, rays and skates expend a large amount of energy producing relatively few, well developed offspring, via oviparous or viviparous (placental/aplacental) strategies. Embryos are maternally nourished either directly (vivipary) or indirectly via the yolk sac (oviparity/aplacental vivipary), thus the chemical composition of tissue formed during embryonic development reflects the diet of the pregnant mother. While this provides a potentially exciting opportunity to study the movements of pregnant females via analysis of lens core chemistry, there are several factors that must first be addressed. The duration of maternal nourishment varies considerably between species, with gestation ranging from months to years, thus the relative proportion of the lens representative of maternal nourishment varies according to species, an estimate of which is required in order to isolate lens tissue formed pre-birth from that representative of exogenous feeding in the juvenile life history stage. Where size-at-birth data are available, this information can be

used alongside the growth relationship parameters in order to isolate lens tissue formed pre-birth from lens tissue accreted post-parturition.

In viviparous species, isotopic variability within the pre-birth portion of the lens is likely to reflect changes in diet and/or location of the gestating female. However, in oviparous and aplacental viviparous species, the pre-birth lens tissue reflects maternal nutrient assimilation during egg sac formation. Given that yolk capsules may be laid down prior to the initiation of embryo growth (*Wood, Ketchen & Beamish, 1979*), there is potential for considerable temporal decoupling between maternal feeding fuelling yolk sac production, and the subsequent utilization and incorporation of this material into the tissues of the embryo. Furthermore, in aplacental viviparous species, the potential for pre-birth omnivory in the developing embryo presents an additional complication. Consumption of unfertilized eggs (oophagy) formed at different times may further complicate the issue of temporal decoupling, whilst consumption of un-hatched embryos (embryophagy) could introduce an additional trophic offset, similar to that associated with maternal partitioning.

## CONCLUSIONS

Many researchers have exploited incrementally grown tissues in fishes for aging studies, investigations into stock discrimination, and isotope ecology. Here we show that eye lens tissue also serves as a repository for stable isotope-derived information relating to the whole life-history of an individual, allowing retrospective investigation of trophic history and habitat use in both teleosts and chondrichthyans. Stable isotope analyses of fish eye lenses can provide information on movements and trophic dynamics of otherwise inaccessible species. The structure and mechanical properties of the fish lens differs markedly between species, however, and can present problems associated with handling. It is therefore important to assess lens morphology on a case-by-case basis, and to develop handling and preparation protocols that can be adapted accordingly to suit different species. Lenses are relatively porous materials, susceptible to contamination by resin-based embedding media. There may be scope for further testing of different embedding media with preferably lower permeability, or solvent-based approaches for removing resin from cut sections. In the meantime, we suggest that chemical-free handling protocols such as 'peeling' the lens are preferable, despite a reduction in control over the sampling interval and associated uncertainty in the time represented by each sample. Further research is also required in order to reliably interpret isotopic compositions of lens tissue formed *in utero* in elasmobranchs, or in other viviparous fishes. Sharks, rays and skates require special attention, and given the range of reproductive strategies, the associated maternal nourishment, and the variable gestation period, each species should be addressed on a case-by-case basis.

In conclusion, the eye lens represents a promising target for biochemical reconstruction of life-long diet and spatial ecology in fishes, but interpretation of lens isotope data, particularly in the pre-birth portion of the lens, requires care and detailed consideration of the relative timescales associated with tissue formation.

## ACKNOWLEDGEMENTS

The authors would like to extend thanks to Crista Bank (The School for Marine Science and Technology, at the University of Massachusetts) and staff at Cefas (Lowestoft) for providing samples. We thank Jason West, Orian Tzadik and an anonymous reviewer for helpful and constructive comments on the manuscript.

### Funding

This study forms part of Katie Quaeck-Davies' PhD studies, funded by the Natural Environment Research Council (NERC) and the Centre for Environment, Fisheries and Aquatic Science (Cefas) (Cefas Seedcorn Number DP308, "Novel Utilisation of elasmobranch biomaterials"). Biological sampling was supported by Defra funded contracts: MB5201 National Evaluation of Populations of Threatened and Uncertain Elasmobranch Stocks (NEPTUNE). MF047 Fisheries Science Partnership 2011–2012: Spurdog, porbeagle and common skate bycatch and discard reduction, and by the Cornish Fish Producers Organisation (CFPO) and vessels supporting the above programmes within the South West commercial net fisheries. Stable isotope analyses were additionally supported by NERC LSMSF grants EK241-12/14 and EK254-09/15. The funders had no role in study design, data collection and analysis, decision to publish, or preparation of the manuscript.

### Grant Disclosures

The following grant information was disclosed by the authors:
Natural Environment Research Council (NERC) and the Centre for Environment.
Fisheries and Aquatic Science (Cefas).
National Evaluation of Populations of Threatened and Uncertain Elasmobranch Stocks (NEPTUNE): MB5201.
Fisheries Science Partnership 2011–2012: MF047.
Cornish Fish Producers Organisation (CFPO).
NERC LSMSF: EK241-12/14 and EK254-09/15.

### Competing Interests

The authors declare there are no competing interests.

### Author Contributions

- Katie Quaeck-Davies conceived and designed the experiments, performed the experiments, analyzed the data, prepared figures and/or tables, authored or reviewed drafts of the paper, approved the final draft.
- Victoria A. Bendall and Stuart Hetherington contributed reagents/materials/analysis tools, authored or reviewed drafts of the paper, approved the final draft.
- Kirsteen M. MacKenzie conceived and designed the experiments, authored or reviewed drafts of the paper, approved the final draft.

- Jason Newton performed the experiments, contributed reagents/materials/analysis tools, authored or reviewed drafts of the paper, approved the final draft.
- Clive N. Trueman conceived and designed the experiments, analyzed the data, prepared figures and/or tables, authored or reviewed drafts of the paper, approved the final draft.

## Animal Ethics

The following information was supplied relating to ethical approvals (i.e., approving body and any reference numbers):

All samples were recovered from dead specimens obtained from fishing operations. Deep water fish were recovered from fisheries research cruises operating under governmental licence (Marine Scotland or Cefas). *L. nasus* ($n = 30$) lenses and associated body size data were collected from sharks incidentally by-caught by off-shore commercial gill-net fisheries within the Celtic Sea between 2011 and 2014, landed under dispensation in association with dedicated Cefas-led scientific fishery studies (*Bendall et al., 2012*; *Ellis et al., 2016*). Lenses and allometric data were recovered from *S. acanthias* ($n = 101$) incidentally by-caught aboard inshore commercial long line fisheries in southern North Sea in 2013. As we are using commercially caught fish and not performing procedures on live animals, we did not need IACUC approval.

## Data Availability

All raw stable isotope data are provided as Supplementary Files, together with the R code used to produce the figures.

## Supplemental Information

Supplemental information for this article can be found online at http://dx.doi.org/10.7717/peerj.4883#supplemental-information.

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
