# Peer review of "Teleost and elasmobranch eye lenses as a target for life-history stable isotope analyses"

_PeerJ, doi:10.7717/peerj.4883_

## Round 0.1 · original submission · Minor Revisions

Both reviewers have provided substantial suggestions that should be considered in the revision. If you disagree with a reviewer, be sure to state why in your response.

Reviewer 1 ·

Basic reporting

The paper is well written in clear and unambiguous English, although some terms may be unfamiliar to the non-specialist and could benefit from brief definition.

All references in the reference list are cited in the text although I found 8 references in the text that are not in the reference list: Schloesser, Hanson, Estrada, Kroger, Matthiessen, Bron, Lynnerup, Kjeldsen. In some parts of the text, Wallace et al (2014) is referred to as Wallace (2014) - lines 68 and 519.

The Introduction provides a very good review into the topic, particularly for this who are unfamiliar with the field of vertebrate eye structure, as the authors summarise very clearly the relevant detail on eye structure, how the lens is formed and why it is a suitable repository of isotopic data. What is less clear is how the aims of the study (lines 212 - 218) relate to, and lead on from the Introduction. In the present draft they stand alone from the preceding Introduction. I was left asking myself 'why' after reading these points. Please provide some background detail/explanation

In my opinion there are some statements in the text that require supporting references; e.g. lines 155-156, 157, 183-184 (in fact the whole paragraph on elasmobranchs line 183 and following) could do with better referencing), 224, 284 and line 565. In lines 52-56, rather than pick a couple of select references it might be better to cite some recent reviews covering larger range of studies?

The paper is well structured with excellent figures that present the data very clearly with appropriate legends of sufficient detail.

I think that the authors should structure the methods/Results sections so it is clearer which samples are being used for which analyses. This is especially so with the stable isotope analysis as the text reads as if measurements were obtained for all species (by my reading of line 275 and following) when this is not the case.

Line 315 - provide statistical support for the statement that standard runs between labs were similar - presumably data were compared?

Experimental design

It is clear as one reads the paper that the sampling design for this study was opportunistic based on various species caught on various cruises and archived and the authors should acknowledge that. I am sure if they were planning this work again they would think about sample sizes and size ranges and redesign their sampling. I an not criticising this as a flaw to the study but I think that this should be acknowledged. Also I think that some 'signpost' statements would be useful. For example, explicitly stating although sample sizes are extremely small in the lens morphology section of the paper, the aim was to compare lens structure across a range of fish taxa and so the data collected provides a good indication of variability in lens structure and permeability (especially as I am guessing that nobody else has ever done this).

As I mention in the previous section, the research questions defined (lines 212-218) need better defining as they stand alone from the Introduction. It is clear that this study fills an identified knowledge gap and is the first step in developing a refined method to use the fish eye lens as a tool to monitor trophic geography and movement patterns.

The Methods provide sufficient detail.

Validity of the findings

This study provides a very useful advance in the development of the fish eye lens as a tool to monitor trophic geography and movement patterns - it is novel and contains valuable data whose publication would help develop this recent field of research.

Due to samples sizes and the questions being asked, some of the results are descriptive and/or lack statistical analysis, however I do not see this as an issue given the novelty of the subject, and the absence of any other data for this topic.

Additional comments

This is a nice first step in trying to develop methods to use the fish eye lends as a recorder. The limitations of the data and approach need to be fully recognised

·

Basic reporting

The authors present a well written and informative manuscript on a topic of high importance. While the authors need to address several minor edits as listed below, a revised version of this manuscript will add a timely and necessary contribution to the literature considering the relatively recent interest in the use of eye lenses to study life history characteristics in fishes. There are several editorial concerns throughout, which are listed here for the attention of the authors:
1) The term “fish” should be used when referencing a single species, whereas the term “fishes” should be used when referencing multiple species. This needs to be addressed throughout the manuscript and changed accordingly.
2) The term “between” should be used for comparisons between two subjects. The term “among” should be used when addressing three or more subjects. This needs to be addressed throughout the manuscript and changed accordingly.
3) Several assumptions and conclusions throughout the manuscript need to be qualified with references, data, or an explanation by the authors. In several examples, the authors need to allow for alternative explanations as well as their own. Individual instances are listed in the following sections.

Further, several specific issues are listed for consideration as well:
1) Line 22: This language is misleading. The archives that are being referenced are actually of elements/minerals/isotopes/etc… I recommend rewriting this sentence to clarify that the archives of the analytes represent behavior and physiology.
2) Line 24: Please consider removing the comma after the word “unfortunately.”
3) Line 45: This sentence makes it sound like the insights are only derivable in marine animals. This is obviously not the case, so I would suggest either clarifying, or re-writing in a non-ambiguous manner.
4) Lines 46 – 48: Please consider adding additional information regarding other positive components of this method.
5) Lines 51 – 56: As written, this sounds like these are the only examples that exist. I suggest either a thorough review of all examples, or clarifying language to make it clear that these are only several examples.
6) Line 57: I recommend clarifying that the isotopic analysis of the organic component in the otolith in this study was not derived as a chronological recorder. As written, this is not clear.
7) Lines 49 – 64: This entire paragraph reads like a fact sheet instead of a story that has a narrative throughout. Please consider re-writing the paragraph to highlight important components in a logical and clear manner.
8) Line 68: I recommend citing Tzadik et al. 2017 that offers a review on this topic.
9) Line 81: Please consider qualifying this sentence with a logical transition. As written, the discussion of the cornea does not flow logically from the previous statement.
10) Line 100: Please consider explaining further how growth rate varies, and whether the reference is to the course of an individual’s lifetime, or whether this is a comment that refers across individuals.
11) Lines 110 – 111: This is very confusing. Several sentences earlier, the authors declared that growth rates are variable. I recommend clarifying this point.
12) Lines 155 – 156: Is this the opinion of the author? Can this statement be qualified? Please consider adding a reference, or other reasoning to qualify this sentence.
13) Line 187: I recommend adding a reference or additional qualifier to this sentence.
14) Lines 195 – 196: This sentence along with the previous one make it sound like the authors are only referring to pelagic fishes. I would recommend clarifying whether this is in fact the case, or not.
15) Line 230: Please consider changing “of the west Scotland…” to “off the west Scotland…”
16) Line 711: Please consider clarifying the page numbers for this citation.

Experimental design

The authors adhered to a strict protocol during sample preparation and present their results in a clear and scientifically-responsible manner. Minor edits are suggested below:

1) Lines 246 – 254: I recommend qualifying the assumption that water temperature will not affect metabolic processes, including mineral/protein deposition within the eye lens. In addition, the assumption that small sample sizes are representative should be qualified. These assumptions should be qualified in the discussion section as opposed to here.

Validity of the findings

The findings in this study are valid and well-supported. However, there are several instances where the authors fail to recognize alternative explanations for their findings. Minor edits are presented below:
1) Line 366: Is lens porosity the only explanation? If so, please clarify why, or cite previous studies.
2) Line 381: Can the authors please explain how the A and B terms were derived? Where they derived pre-analysis? Were they derived by weight ratios? I recommend clarifying this point.
3) Lines 382 – 384: Please consider validating this assumption. Could this assumption present a problem in the analysis of fishes that live in highly contaminated waters? I agree that the assumption is valid for this study, but I would recommend validating the statement to clarify that the values were obvious for this study, but that future studies should at least consider the possibility that the division will not be so clear.
4) Lines 433 – 434: Please consider quantifying this variability. The figure does not clearly qualify the statement.
5) Line 441: Please consider quantifying this variability for consistency purposes.
6) Lines 464 – 465: I recommend adding that this has implications for age as well if a length-age relationship has been previously determined.
7) Lines 521 – 522: Please consider qualifying this statement with a citation or further examples.
8) Line 546: Please consider qualifying why this is considered an “isotopically constant nutrient source.” I can imagine very erratic behavior from a pregnant mother that may lead to altered feeding behavior.
9) Line 571 – 573: Have the authors considered siblicide as another explanation or isotopic variability?
10) Lines 588 – 589: Could siblicide be a consideration here as well?

Additional comments

Overall, the material is presented in a concise yet comprehensive manner. The figures, tables, and supplementary materials are all straight forward and easily understood. The authors do a good job of reporting their results, stating their analysis, yet not over-reaching in their interpretations of their findings.

---

## Round 0.2 · accepted · Accept

The authors have sufficiently responded to the comments and suggestions from the reviewers.

#